# "It just feels right": Perceptions of the effects of community connectedness among trans individuals

**Jessamyn Bowling** [1]*, **Jordan Barker**[2], **Laura H. Gunn** [1,3], **Tatim Lace**[4]

**1** Department of Public Health Sciences, University of North Carolina at Charlotte, Charlotte, NC, United States of America, **2** Department of Epidemiology, Rollins School of Public Health, Emory University, Atlanta, GA, United States of America, **3** School of Public Health, Imperial College London, London, United Kingdom, **4** Independent Researcher, Charlotte, NC, United States of America

* Jessamyn.bowling@uncc.edu

**Data Availability Statement:** All 5 files are available from the Open ICPSR database at https://www.openicpsr.org/openicpsr/project/117143/version/V1/view.

## Abstract

Trans individuals (e.g. trans men and women, non-binary, gender fluid) are at higher risk for mental health concerns, in part due to marginalization. Previous work has documented the effects of social support and community engagement on health outcomes. However, individuals' perceptions of community engagement effects may point to opportunities for intervention. This mixed methods study examines trans individuals' perceived effects of participating in trans communities on health outcomes. Semi-structured in-person interviews were conducted with 20 individuals, and 60 individuals participated in cross-sectional online surveys. Perceived discrimination had a strong negative association with self-assessed mental health. Effects of participation included a) positive: contextualization and normalization of experience, self-appraisal, safety, and relief; and b) neutral/negative: energy drain and stigmatization. Effects of not participating included loneliness, depression, isolation, hiding identity, and losing resources. Both discrimination and non-participation in trans communities had negative effects on mental health. Though community participation is often discussed as positive for marginalized populations, it may be important to include possible negative effects (such as energy drain) in research.

## Introduction

Trans communities remain an understudied area of research, especially in relation to the effects of community participation and engagement [1]. Trans individuals refer to those whose gender identity differs from societal expectations based on sex assigned at birth (e.g. trans men and women, gender fluid, non-binary). Previous studies of trans individuals are typically focused on transition-related and HIV-related care [2]. Research shows that these individuals are at a higher risk of negative mental health outcomes. These effects may be explained through the minority stress framework, which describes how stress of marginalization within *cisnormative* (prioritizing and normalizing cisgender, or having one's gender correspond to their sex assigned at birth) societies can lead to negative health outcomes [3]. Another

**Funding:** This study was funded by the University of North Carolina at Charlotte (JB). The funders had no role in study design, data collection and analysis, decision to publish, or preparation of the manuscript.

**Competing interests:** The authors have declared that no competing interests exist.

framework to describe the mental health effects of marginalization of trans people is *cisgenderism* which "delegitimizes people's understanding of their genders and bodies" [4, 5]. Discrimination has been found to affect health among trans individuals, both in quantitative (including mental health [6–11] and risk behaviors [12, 13]) as well as qualitative studies (including mental health [14, 15] and health care access [16]). While trans individuals are becoming more visible within society through media and popular culture, these communities continue to experience specific health disparities and stressors. These adversities can prevent trans individuals from achieving their optimal quality of life.

The transgender population experiences stigma and adverse mental and physical health effects at higher rates than their cisgender peers [17, 18]. Specifically, suicide, anxiety, depression, and substance use statistics among transgender individuals are disproportionately high compared to the national average. As of 2015, 48% of respondents reported serious suicidal thoughts in the last 12 months, compared to 4% of the US general population [19]. One-quarter of respondents also reported abusing substances within the last month, as compared to 8% of the US population [19]. Furthermore, depression (44.1%) and anxiety (33.2%) were reported at higher rates than the national averages of 7.1% and 19.1%, respectively [20].

In order to combat these negative health outcomes among trans individuals, protective factors such as community engagement and social support can mitigate negative health outcomes and provide the ability to cope with the burdens of stress and prejudice that trans individuals may experience daily from family, coworkers, and friends [17, 18]. Social support can be categorized in the following ways: emotional, instrumental, appraisal, and informational [21]. Online communities constitute one component of social support. Cipolletta [22] suggests that online communication with other trans individuals can provide safety for disclosing intimate thoughts and advice regarding the transition process while allowing the user to protect their identity. Participants within this study cited experience sharing, asking and offering advice, and building relationships as three of the top motivations for engaging in an online community. Previous research has reported linkages between mental health status and social support among trans people. Lack of support from family and friends, higher risk of cyberbullying, and personal violence are all correlated with negative mental health outcomes for trans individuals [23]. Family can provide fundamental social support that is pivotal to mental health and self-esteem. A lack of familial support and maltreatment from the family unit increased the chances of the trans individual becoming homeless by nearly four times [24]. Self-esteem was also shown to be linked to familial acceptance, with lower familial acceptance associated with, for example, a decreased chance of future gender affirmation medical procedures for those individuals desiring them. This study suggested the implementation of social support networks to create higher self-esteem ratings within the transgender and trans community.

Historically, lesbian, gay, bisexual, trans, and queer (LGBTQ+) individuals have formed or engaged in groups for support, leading and advocating for change, and to celebrate diversity across sexuality and gender [25–28]. This study examines community engagement broadly, including community participation and connectedness. Community connectedness is defined as "the strength of an individual's affiliation with a group to create a mutually influential relationship and sense of belonging" [29–32]. Pflum's [18] study of transfemme individuals (those who were not assigned female at birth and present as women or feminine, see [33] for more on this identity) identified that increasing community connectedness and seeking social support are linked to a decrease in adverse mental health effects. In building a connection with a community, trans individuals may seek support for a variety of reasons, and not always from trans-specific communities. One study had participants describe their supportive groups and how these groups aided them. Participants who felt connected to a variety of communities, such as LGBTQ+, religious, and ethnic groups, reported that these groups encourage them when they

are feeling "overwhelmed and less resilient," and having a community to rely on allows them visibility within society [34].

Though definitions of communities vary depending on context, we use a broadly inclusive definition of a shared similarity between group members [35]. Community resilience is a recent shift in understanding various socioecological levels of resilience from individuals and families. Although subjective community experiences among trans individuals have not been described in academic literature, individual experiences of resilience strategies have [34]. Though trans individuals are not monolithic or a single community [36], it may be relevant to examine groups as an important facet of improving health outcomes among this population. The subjective experience of engaging in communities may influence the success of community resilience; in previous research on community resilience, individuals' sense of community could be used to predict disaster preparedness [37]. Therefore, understanding individuals' perceptions of community engagement may reveal opportunities for intervention and community approaches.

The purpose of this study is to examine trans individuals' perceived effects on health outcomes of participating or not in trans communities. We use survey data to examine the effects of discrimination and interview data to examine the effects of trans community participation. As the study is based in a large urban area in the South (Charlotte, NC), it may be important to describe some aspects of the general context. The racial and ethnic breakdown of Charlotte, NC is approximately 42% white, 35% black/African American, 14% Hispanic/Latinx, 6% Asian, 3% other [38]. The city is largely segregated socially by race and ethnicity, with different organizations focused on different groups. There is a prominent and active LGBTQ+ youth organization in the city but no LGBTQ+ support organization for individuals over 21 years of age. Our community partner for this study, Transcend, is a predominantly white local trans organization that holds bi-weekly support meetings both for trans individuals as well as family/partners/friends and has a clothing closet. Other trans spaces in the city at the time of the study include a comic book store and café, a trans women's support group, HIV service organizations, specific nights at various nightclubs and bars, and annual Pride events.

## Materials and methods

This community-based participatory research study used semi-structured interviews (n = 20) and an online survey (n = 60). The interview guide and survey were developed by study authors JB and TL, based on existing literature and preliminary conversations with the community partner, a trans support organization. The community partner's leadership (two individuals) reviewed the guide and survey and made edits; due to time and financial limitations, we did not conduct a pilot study. Eligibility criteria for both the survey and interviews included being at least 18 years of age, identifying as other than cisgender (e.g. trans, genderqueer, etc.), living in the Charlotte, NC region, and an English speaker. Participants were recruited simultaneously for both methods through word of mouth and social media posts in LGBTQ+ and trans groups. Participants were selected if they met eligibility criteria. All protocols and procedures were approved by the Institutional Review Board of the University of North Carolina at Charlotte.

Potential interview participants were directed to an online screening questionnaire after indicating informed consent electronically. Interviews took place in private rooms within an LGBTQ+ community organization or on the phone for those in extenuating circumstances (e.g. lacking transportation; n = 2). The interviewer was a trans individual familiar with the community, identified by the community partner. Interviews lasted approximately 45 minutes and focused on experiences of community engagement and local trans organizations. Example

questions included, *"What are the effects on your daily life when you participate in gender diverse groups/services?"* and *"What are the effects on your daily life when you don't participate in gender diverse groups/services?"* Participants selected their own pseudonyms and these are used in this paper. Participants received a $20 gift card for their participation. All interviews were audio recorded and transcribed verbatim.

Interview transcripts were coded using inductive thematic analyses [39] with Dedoose online qualitative analysis software [40]. Common ideas were grouped together to form "themes." A codebook was created based on the interview guide and then augmented during initial analyses. Two different trained coders coded each interview such that each interview was coded twice. Reliability was confirmed among the coding team using Dedoose's test function, with any codes corresponding to a Kappa of less than 0.80 discussed and refined until consensus was reached. Coders used "memo"ing, or digital notes tied to excerpts or interviews, to track their assumptions and biases while coding [41]. Initial analyses were presented as a summary report to the community partner's leadership for verbal feedback during an in-person meeting.

Surveys were conducted using Qualtrics online software. Survey questions focused on physical and mental health, discrimination, community connectedness, strengths and weaknesses of the community partner organization, and effects of participating with trans communities. Participants received $15 Amazon gift cards for completing the survey.

In addition to the qualitative analysis, a quantitative assessment of the influence of discrimination on both physical and mental health was performed. Survey data was mapped in ordered form so that higher values of the ordered variable were associated with higher levels of the underlying latent construct. A description of this mapping is found in Table 1. A latent variable modeling approach was designed to extract information from the categorical (ordered) survey responses into latent constructs. Associations between latent constructs were explored using the R package *lavaan* [42]. The latent variable *discrimination* was constructed from seven ordered responses, defined on a 6-point Likert scale, relating to the topic, and it was used as a latent covariate in regression against self-assessment measures of physical and mental health, defined on a 5-point Likert scale, provided by the respondents.

**Table 1. Survey questions regarding discrimination and physical and mental health, as well as the mapping to ordered variables.**

| ID | Survey Question | Latent Measure | Survey Responses & Ordered Response Mapping |
|----|-----------------|----------------|---------------------------------------------|
| Q1 | In your day-to-day life how often have any of the following things happened to you because of your gender identity?—You are treated with less courtesy or respect than other people. | Discrimination | Never = 1<br>Less than once a year = 2<br>A few times a year = 3<br>A few times a month = 4<br>At least once a week = 5<br>Almost every day = 6 |
| Q2 | In your day-to-day life how often have any of the following things happened to you because of your gender identity?—You receive poorer service than other people at restaurants or stores. | | |
| Q3 | In your day-to-day life how often have any of the following things happened to you because of your gender identity?—People act as if they think you are not smart. | | |
| Q4 | In your day-to-day life how often have any of the following things happened to you because of your gender identity?—People act as if they are afraid of you. | | |
| Q5 | In your day-to-day life how often have any of the following things happened to you because of your gender identity?—You are threatened or harassed. | | |
| Q6 | In your day-to-day life how often have any of the following things happened to you because of your gender identity?—People act as if they're better than you are. | | |
| Q7 | In your day-to-day life how often have any of the following things happened to you because of your gender identity?—You are called names or insulted. | | |
| Q8 | Thinking about the last 6 months, how would you rate your PHYSICAL health? | Physical Health | Poor = 1<br>Fair = 2<br>Average = 3<br>Good = 4<br>Excellent = 5 |
| Q9 | Thinking about the last 6 months, how would you rate your MENTAL health? | Mental Health | |

## Results

A general overview of the themes and subthemes that emerged from this study is included in Table 2.

### Effects of discrimination on mental and physical health

Fig 1 shows the graphical representation of the latent variable modeling assessing the impact of perceived discrimination on mental and physical health. A summary of results from this latent variable model is provided in Table 3. All measures of discrimination reported in Table 1, with the first measure set to 1 for identifiability purposes, provided similar, statistically significant contributions to the latent construct, as reflected in Table 3. Negative values of the regression coefficients between perceived discrimination and self-assessed mental and physical health indicate that higher levels of perceived discrimination are associated with lower levels of self-assessed mental and physical health. While both estimated regression coefficients follow that negative pattern, only the one relating to the relationship between perceived discrimination and self-assessed mental health was statistically significant ($\beta$ = -0.519; p<0.0001).

### General connectedness

Interview participants in our study were predominantly white (n = 18, 90%), 23–30 years of age (n = 7, 35%), and trans women/feminine (n = 8, 40%) (see Table 4 for participant demographic information). In general, some participants reported it was difficult to discern the effects of connecting with trans groups because they had adjusted to a negative and stigmatizing normality. "*I got so used to normal day-to-day life without a lot of exposure to other trans people. . .So, really it was kind of like living life. . . as a regular person*" (S, 51–60 years, male (female to male; FTM)). Similarly, others had trouble describing effects because there was no distinction between when they felt connected or not to trans groups; these individuals had friends and family who were trans, and they were in nearly constant contact with trans

**Table 2. Themes of the effects of discrimination and community connectedness from both qualitative and quantitative methods.**

| Theme | Subthemes |
|---|---|
| Perceived effects of discrimination on health | Increased levels of discrimination significantly associated with mental health |
| General trans community connectedness | Difficult to discern effect on life because: |
| | 1) participants adjusted to stigma in daily life, and/or 2) there's no distinction of connecting (or not) to trans community |
| | Level of community engagement varied |
| Effect of trans community connection: | Feel better |
| | Social support |
| *Positive* | Normalized and validated experiences |
| | Freedom of expression |
| | Safety and relief from risk and stress of life |
| *Neutral* | Drains energy while also regenerative |
| *Negative* | Hearing others' negative experiences is triggering |
| | Being in group of trans individuals can increase perceived stigma |
| Effects of not connecting with trans community | Feel lonely and guarded |
| | Begin to isolate socially |
| | Difficulty finding resources |
| | Work/daily lives become more encompassing |

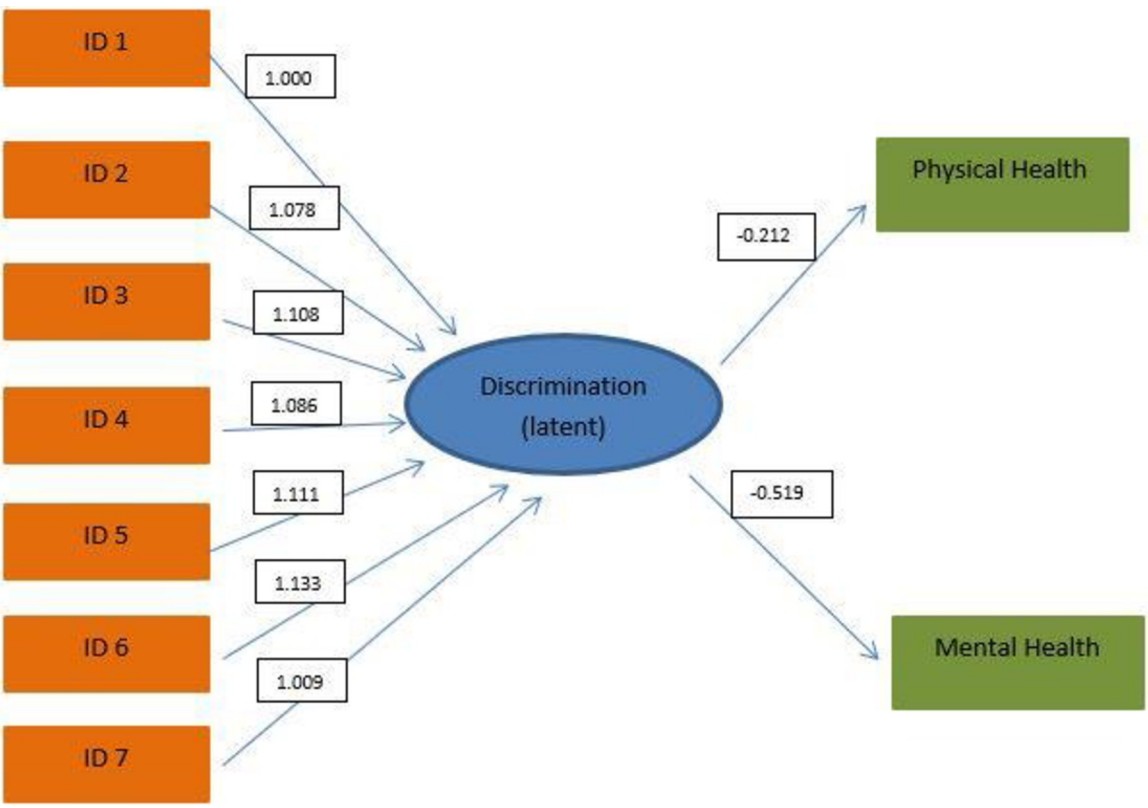

**Fig 1. Graphical representation of the latent variable model, with numbers 1–9 representing the IDs listed in Table 1 regarding the Likert-scale questions in the questionnaire.** Boxed numbers represent the model estimates, as listed in Table 3.

communities. "*I'm really fortunate that my husband is trans. [. . .] and I have a trans roommate, and I have another trans partner; so, most days I participate in trans groups, I feel like.*" (Max, 23–30 years, non-binary transmasculine)

Participants' level of engagement with trans communities varied, with some more deeply embedded in activism and advocacy and others hardly involved with other trans individuals. Amy (51–60 years, male to female (MTF) trans) spent most of her time disconnected from trans communities: "*Considering that I am not out professionally at all, assuming that I'm not*

**Table 3. Impact of perceived discrimination on self-assessed mental and physical health.**

| Latent Variable Construction | Estimate | SD | z-value | p-value |
|---|---|---|---|---|
| Discrimination Q1(Fixed estimate to provide scale) | 1.000 | | | |
| Discrimination Q2 | 1.078 | 0.072 | 14.990 | <0.0001 |
| Discrimination Q3 | 1.108 | 0.061 | 18.193 | <0.0001 |
| Discrimination Q4 | 1.086 | 0.058 | 18.603 | <0.0001 |
| Discrimination Q5 | 1.111 | 0.070 | 15.814 | <0.0001 |
| Discrimination Q6 | 1.133 | 0.073 | 15.519 | <0.0001 |
| Discrimination Q7 | 1.009 | 0.075 | 13.517 | <0.0001 |
| Discrimination (Latent) | 0.630 | 0.077 | 8.217 | <0.0001 |
| *Regressions*: | | | | |
| Physical Health Q8 ~ Discrimination | -0.212 | 0.151 | -1.407 | 0.159 |
| Mental Health Q9 ~ Discrimination | -0.519 | 0.130 | -3.987 | <0.0001 |

**Table 4. Interview participant demographic information.**

| Pseudonym | Pronouns | Gender ID | Race/ Ethnicity | Age Range (Years) |
|---|---|---|---|---|
| Gwen | she/her | transgender woman | white | 23–30 |
| Susan | she/her | transgender female | white | 31–40 |
| Britney Spears | she/her | trans female | black | 23–30 |
| Amy | she/her | MTF trans | white | 51–60 |
| Mina | she/her | trans woman | white | 23–30 |
| Miranda | she/her | trans woman | white | 31–40 |
| Nicole | she/her | trans woman | white | 41–50 |
| Wonder Woman | she/her | elder trans woman | white | 51–60 |
| Lara | she/her | female | white | 41–50 |
| S | he/him | male (FTM) | white | 51–60 |
| Cooper | he/him | male (FTM) | white | 41–50 |
| Jeff | it/its | genderfluid trans man | white | 23–30 |
| Max | they/them | non-binary transmasculine | white | 23–30 |
| Mike | he/him | Changes daily but masculineish sometimes | white | 18–22 |
| Chaucer | they/them | genderqueer/gender fluid | white | 23–30 |
| Elizabeth | she/her | transfeminine nonbinary | white | 41–50 |
| Geo | they/them | non-binary | black | 18–22 |
| Gio | they/them | non-binary/agender | white | 18–22 |
| Jim Henson | they/them | nonbinary | white | 23–30 |
| Scotland | they/them | agender | white | 41–50 |

*working on the weekends, the weekends is all that I have. Weekends and evenings once I get home.*" Amy's disconnection was, in part, tied to her belief that individuals navigate their own paths, but she looked forward to connecting with more trans people through a support group.

## Effects of trans community connection

Many participants described their connection as making them feel better, "*balanced*" (Jim Henson, 23–30 years, nonbinary), or "*bolstering their emotional health*" (Gio, 18–22 years, nonbinary/agender). Part of this was in relation to contextualizing their experiences.

> "*It helps me remember, too, that there are other trans people out there and some that are in much worse situations. In general, my transition was good, my family overall was pretty accepting. Work was pretty good. Didn't lose friends or else they were there at the beginning or kind of phased out.*" (S, 51–60 years, male (FTM))

Social support was one of the most predominant effects of trans communities' connection that participants stated. In addition to having individuals who could understand an asexual identity, Mike (18–22 years, gender changes daily but masculineish sometimes) had identified others who could understand his identities:

> "*I feel like I can text just about anyone at [community organization] at any time and know that I have a friend to back me up. . .I'm going to call it intimacy with my friendship, that there is a certain level of people that I can just vibe with and they understand and I don't have to explain myself every five minutes. They understand. They'll ask me 'What are your pronouns? What is your name today? Who is presenting today?'"*

One facet of social support can be assisting in self-appraisal. Elizabeth describes seeing themselves as a resource, which may be tiring while also affirming.

"*I guess my participation in gender diverse groups or services is very different from someone who relies on them as a resource, because in many cases I am the resource. My phone goes off constantly. There's always somebody popping up in my messenger.*" (Elizabeth, 41–50 years, transfeminine nonbinary)

Their networks were constructed through involvement in trans communities.

Trans community connections normalized and validated participants' experiences. Online communities were helpful for general information finding (such as hormone advice) and for a wide variety of identities.

"*I was really afraid to transition for a long time because I thought that because my sister is trans, because my partner is trans, that it wasn't okay for me to also be trans. And I think that hearing from other trans people, that's another place where the internet portion comes in, because there's not a whole lot of people in the whole world whose sibling is also trans. But if you're going to find them, you will find them online and I'm not the only one and that's pretty encouraging.*" (Jeff, 23–30 years, genderfluid trans man)

The mere fact of others' existence was enough to normalize a wide range of experiences.

"*It was enough confirmation that there were other people, that it wasn't necessarily something that I needed for support. At the same time, I was just transitioning and I just had some chest surgery, so it was kind of a confirmation or affirmation of who I was. . . but something that, 'wow, there are other people like me.'*" (Cooper, 41–50 years, male (FTM))

Similarly, participants described a freedom of expression in their communities not only in identity and appearance but also ideas. "*I just like scrolling through my feed and it's nice. . .I like seeing content that is both body and size positive and trans positive 'cause that's something that I struggle with.*" (Gio, 18–22 years, nonbinary/agender) This was related to participating in both online, as well as virtual, groups (such as a local trans community Discord group, a chat app that allows users to create anonymous profiles).

"*Through the Facebook page, it was always kind of tough. I did some through the messenger there. For some reason, that didn't seem as 'let your hair down' kind of thing. Whereas [with] the Discord server, I jump on and I'm part of any conversation I want to be in and they know me and I know them and it's more of a conversation.*" (Nicole, 41–50 years, trans woman)

They could also learn about others' gender diversity through the full range of identities participating in their communities.

Unsurprisingly, participants often described feeling safety and relief from the risks and stress of their lives outside of trans communities. "*I know that I have something that week where I can comfortably be myself; so, even if I am having one of those terrible days at work or with my parents, then I at least have that relief coming.*" (Chaucer, 23–30 years, genderqueer/ gender fluid) For Max (23–30 years, nonbinary transmasculine), this was relief from constantly having to be the only trans, or even LGBTQ+, person at work or in spaces, "*I'm usually the only trans person in a room and that's exhausting; so, I love being around other trans people who are just living their lives, and I'm like 'Thank God.'*"

Not all of the effects of participation were fully positive. One neutral effect was the drain of social interaction, though trans community participation was also regenerative. "*While I can't come to [community organization] every week because of my social battery,. . .going to gender diverse groups gives me energy a lot. It takes energy to get here, but it gives me energy in return.*" (Mike, 18–22 years, gender changes daily but masculineish sometimes) There were two types of negative effects identified by participants. The most common was that hearing others' similar negative experiences (such as stigma or family trouble) was triggering, and participants sometimes felt exhausted or sad because of the triggers.

> "*For the most part, I am happy around my friends and we love to joke around and talk about different issues or even just bitch about life and some of the stuff that sucks about being trans. Sometimes it can have a negative effect on me where it's more, so bittersweet.*" (Gwen, 23–30 years, transgender woman)

Moreover, action steps in relation to these negative experiences were often not discussed.

> "*I find that when we are just getting together and having fun, I find those like really, really refreshing. And I do want to seek out spaces where trans people are trying to learn how to cope with trauma, but I haven't found a space that does it in a way that I think is actually healthy.*" (Max, 23–30 years, nonbinary transmasculine)

A second negative effect was that being in a group of trans individuals in public can be more stigmatizing than just being by oneself.

> "*All of us being gender diverse, when we are in groups together, it can be like, we go out and we're harassed. Or people give you a microaggressive look, or they stare. And that can be stressful if I'm going out like, 'Do I need to prepare myself? What am I going to do if someone misgenders my friend?'*" (Geo, 18–22 years, nonbinary)

### Effects of not connecting with trans communities

Nearly universally, participants said that when they did not connect with trans communities or groups, they felt lonely. "*I get depressed. I get very—I don't want to say irritable, but I get very sensitive to things.*" (Nicole, 41–50 years, trans woman) Participants also described feeling more guarded. "*I become a little bit [more] guarded. I'm more comfortable and at ease amongst those of my own kind, basically.*" (Wonder Woman, 51–60, elder trans woman)

Participants said that when they felt less connected to trans groups or communities, they began to isolate themselves in general. "*I am just kind of like, more reclusive and lonely 'cause coming to [community organization] right now is one of my main social outlets. So, I would just be kind of a shut in.*" (Mina, 23–30 trans woman) This isolation resulted in the loss of relationships for some, like Susan. "*I kind of lose touch. I isolate. I do forgo and sometimes get way too lost and everything else. . .so, the loss there is just really relationships.*" (Susan, 31–40, transgender female) More specifically, some participants said they began "hiding" in relation to their gender identity, as Nicole (41–50 years, trans woman) says, "*I go back into hiding once I leave here.*" This 'hiding' meant not expressing themselves as fully, or correcting people when they are misgendered. Participants described this as detrimental and grief-causing. Scotland reported feeling "*shitty*" when they are less connected and do less advocacy for themselves.

"*The lack of participating in these groups is what gets me. It irks me, and I know a lot of it is on me, but I always have this kind of itty bitty internal knee jerk when anyone refers to me as 'Mr' or use male pronouns. But part of the reason they do is because I haven't corrected them. And the reason I haven't corrected them is, frankly, I don't want my head broken and I want my contract renewed. . . And that's a shitty way of hiding in the patriarchy.*" (Scotland, 41–50 years, agender)

Related to isolation, participants had trouble finding resources in the community because they were less connected with trusted individuals to whom they could turn for ideas and referrals. S described what would be lost if he was not connected with other trans individuals in the community.

"*[Local groups are] good for local resources, because sometimes you look for resources and either you can't find it in your area or it's dated or the doctor is no longer practicing or has moved. So it's nice here that you have all of that information you can access and it's local and its current.*" (S, 51–60 years, male (FTM))

When they were not as connected to trans communities, participants' work or daily lives became more encompassing. This was often described as problematic because these spaces were stigmatizing or non-affirming.

"*As long as I'm focused on my work, it's what my brain thinks. But not being out at work a lot can cause some grief, you know. Especially with like, going to the restroom, so I don't do the wrong one. [Laughter] Or you go out to lunch with the other coworkers and not being recognized as the woman that I am can sting a little bit.*" (Amy, 51–60 years, MTF trans)

Participants also described a general disconnectedness. This included missing the community conversations, but also a general feeling of something being wrong.

"*I am missing out on part of my community who I should be around. It's like we affirm each other. Like each other's existence and just being around each other. And when I don't have it, something is missing, it's not right.*" (Geo, 18–22 years, nonbinary)

## Discussion

This study provides evidence of positive effects from community connectedness for trans individuals. The vast majority of participants reported that they noticed the effects on their lives when they did or did not participate in trans communities. Participants described effects on their own mental health, but also how community engagement built their social support networks which may be an important aspect of resilience. Previous reviews have documented how community engagement positively affects physical and psychological health [among African American women, 43], self-confidence, self-esteem, sense of personal empowerment, and social relationships [44]. Participants were less likely to describe effects that were prosocial, or focused on helping others; though a few participants mentioned that they were a resource for others, they did not discuss this as a benefit. Other studies have found cultivating empathy and providing support to others as themes of community support among trans individuals [45].

Hendricks and Testa [46] report that disclosing gender identity may facilitate community resource utilization. Given that participants in our study discussed learning about gender

identities and receiving self-appraisal support, they may be navigating their own gender identity. Thus, the barrier of self-disclosure may be problematic for those who most need community. In other words, trans individuals without language to talk about their gender identity may be less able to access community yet in need of resources and support.

Participants in our study described some negative effects of engaging in trans communities, such as being triggered by negative events or emotions that were similar to their own. Other work with trans individuals found competition in groups, of being "trans enough" [45]; participants in the current study did not discuss this, in part perhaps due to the diversity of gender identities present in the group. In community organizing or support, it may be important to build skills around coping with vicarious trauma [occuring from working with individuals with trauma; 47]. Or, as Max discussed, ensuring that trans individuals have different ways of interacting other than sharing negative experiences. As Mike described the energy drain of participating in groups, this may have implications for collective action. Previous work has identified how trans people of color can be drained from activism efforts [25, 48]. This may be tied to findings that higher levels of antitrans discrimination are associated with collective action [49].

Emotional support was the most common benefit from community participation, and reported as a loss when not participating in trans communities. This was described as validation, freedom of expression, and empowerment. Furthermore, participants described informational support in obtaining referrals for local providers or resources; this can also be described as "investing in community knowledge" [45] as individuals continue to share this awareness within and among trans groups.

Matsuno and Israel's [50] transgender resilience intervention model highlights community resilience and intervention opportunities for trans individuals. Group interventions include group therapies and support groups, mentoring programs, and/or family/couples' therapy. As many participants in our study appreciated learning about other trans identities as well as describing being triggered by others' experiences, interventions that offer distraction, are socially focused, or non-therapy focused (such as outings to restaurants or game nights) may be important interventions for positively supporting trans individuals. Further, Case and Hunter [51] describe the power of "counterspaces" in supporting positive adaptation for marginalized individuals by "challenging deficit-oriented narratives." We find evidence of all of the counterspace framework domains: *narrative identity work* in which participants described learning about others' identities, *acts of resistance* in which participants described the value of freedom of expression that are not sanctioned in larger society, and *direct relational transactions* in which individuals offer social support, safety, and security.

Trans community interventions may expand outside of in-person modalities, including virtual and online. As Metthe [45] points out, online communities are a key part of trans social support. Though we did not specifically ask about online networks, many participants in this study differentiated the benefits of in-person communities (deeper connections, being known), virtual communities such as Discord chats (easy access, anonymity, local knowledge), and online communities (wide range of individuals, numerous communities). In localities where individuals have limited transportation access or with widespread geographic areas, virtual and online communities may become more salient.

In addition to the qualitative analysis of connectedness, we identified a strong quantitative association between perceived discrimination and self-assessed mental health; though, a similar statistically significant result was not confirmed for the association between perceived discrimination and physical health. This relationship between discrimination and mental health is reported in other samples of trans individuals [5, 52]. This may be a factor of both a potentially weaker negative association between perceived discrimination and physical health, as

well as the smaller sample size of the study. Within the qualitative results, we see participants more focused on mental health impacts than physical health; the one subtheme that connected with physical health was that participants had difficulty finding resources (including healthcare providers) when they were less connected to the trans community. The impact on self-assessed mental health of any perceived discrimination, and the potential impact (primary or secondary) of this association on self-assessed physical health, should grant further study in this area. These results serve a dual purpose: (1) they provide further evidence of the hypothesis that discrimination has a negative impact on mental health; and (2) they can serve as a pilot or feasibility study, providing measurable outcomes that can help assess statistical power in future studies on the impact of discrimination on both physical and mental health within the trans community.

## Strengths and limitations

This study used community-based participatory research approaches and a diverse research team, which may have increased the validity of our findings. However, as most participants were connected to the community partner, our findings may reflect experiences of trans individuals who are more involved in trans communities. Because all in-person interviews were conducted at the community partner location, this may have excluded the experiences of those uncomfortable with the organization. Individuals who are less connected may not notice or experience the same benefits of participating in trans communities. Interview participants were mostly younger and white, in part due to our community partnership and recruitment through social media.

## Supporting information

**S1 File. Interview questions.**
(DOCX)

## Acknowledgments

We appreciate Transcend's partnership in this community-based participatory research project.

## Author Contributions

**Conceptualization:** Jessamyn Bowling, Tatim Lace.

**Data curation:** Jessamyn Bowling.

**Formal analysis:** Jessamyn Bowling, Jordan Barker, Laura H. Gunn, Tatim Lace.

**Funding acquisition:** Jessamyn Bowling.

**Investigation:** Jessamyn Bowling, Tatim Lace.

**Methodology:** Jessamyn Bowling.

**Project administration:** Jessamyn Bowling.

**Writing – original draft:** Jessamyn Bowling, Jordan Barker, Laura H. Gunn.

**Writing – review & editing:** Jessamyn Bowling, Jordan Barker, Laura H. Gunn, Tatim Lace.

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
