## [Decision Letter · Decision Letter 0]

12 May 2020

PONE-D-20-00790

“It just feels right”: Perceptions of the effects of community connectedness among gender diverse individuals

PLOS ONE

Dear Dr. Bowling,

Thank you for submitting your manuscript to PLOS ONE. After careful consideration, we feel that it has merit but does not fully meet PLOS ONE’s publication criteria as it currently stands. Therefore, we invite you to submit a revised version of the manuscript that addresses the points raised during the review process.

Please respond to each comment from the two reviewers, including revisions made in response to those comments, in your next draft.

We would appreciate receiving your revised manuscript by Jun 26 2020 11:59PM. To enhance the reproducibility of your results, we recommend that if applicable you deposit your laboratory protocols in protocols.io, where a protocol can be assigned its own identifier (DOI) such that it can be cited independently in the future. For instructions see: http://journals.plos.org/plosone/s/submission-guidelines#loc-laboratory-protocols

We look forward to receiving your revised manuscript.

Kind regards,

Amy Michelle DeBaets, PhD

Academic Editor

PLOS ONE

Journal Requirements:

2. When reporting the results of qualitative research, we suggest consulting the COREQ guidelines: http://intqhc.oxfordjournals.org/content/19/6/349. In this case, please consider including more information on how participants were selected; if a pilot study was tested; how data was coded; if bias issues were considered.

3. Please endure that the statements made at lines 109 to 111 are supported by references. Moreover, please state the name of the community partner mentioned at line 112.

4. Please include additional information regarding the survey or questionnaire used in the study and ensure that you have provided sufficient details that others could replicate the analyses. For instance, if you developed a questionnaire as part of this study and it is not under a copyright more restrictive than CC-BY, please include a copy, in both the original language and English, as Supporting Information. Moreover, please include more details on how the questionnaire was pre-tested, and whether it was validated.

Additional Editor Comments (if provided):

Reviewers' comments:

Reviewer's Responses to Questions

**Comments to the Author**

1. Is the manuscript technically sound, and do the data support the conclusions?

Reviewer #1: Yes

Reviewer #2: Partly

2. Has the statistical analysis been performed appropriately and rigorously? 

Reviewer #1: Yes

Reviewer #2: Yes

3. Have the authors made all data underlying the findings in their manuscript fully available?

Reviewer #1: No

Reviewer #2: Yes

4. Is the manuscript presented in an intelligible fashion and written in standard English?

Reviewer #1: Yes

Reviewer #2: No

5. Review Comments to the Author

Reviewer #1: Comments to author

The abstract summarizes the study in a comprehensive manner.

Introduction:

Good review of the pertinent literature.

Line 40 I would remove “to be” after “remains”

The sentence starting by “Self-esteem” on line 76 presents pertinent information but may be perceived as too binary in its orientation. The sentence recounts the results from a study performed in 2018 and speaks to lack of familial acceptance decreasing the chance of completing sex reassignment surgery. Not all gender diverse individuals choose to undergo surgery and in fact many don’t want it. Therefore, completing surgery is not a required part of the self definition of one’s gender. The sentence appears to indicate the contrary. The authors may want to rethink how they present this data.

Material and Methods

In my opinion this is the part of the manuscript that needs the most work. The authors have used a mixed method approach. The quantitative portion of the study uses a questionnaire that was devised de novo and which has not been validated. The sentence starting by “The initial” on line 119 briefly states that the interview guide and survey were informed by “the community partner”. This is all that is said about how these two instruments were developed. We need more information on that process. E.g. By whom and how were the questions used in the interview guide and survey originally drafted? Was this based on existing literature? What does “informed by the community partner” mean exactly? Were only one or two people consulted, was it a committee? How were these individuals chosen to give their opinion? Did they suggest questions, some or all? Did they read and say, yes that sounds fine”

As stated in line 127 and 128 the interviews occurred at a LGBTQ+ community organization. This may have led to individuals not wanting to participate as they may not have wanted to be seen entering such an area. This is not mentioned in the limitations and it should.

The authors present a good description of their coding process, but some items need to be clarified. In Line 136 it is mentioned that “Each interview was coded twice by a trained coder”. This is not clear to me. Does this mean that the interview was coded by two different coders? Or twice by the same coder? As it is written it seems to be the latter. Is this right? They also mention in line 138 that they presented the initial analyses to the community partner for feedback. How as this done, to how many people, where they the same as those that participated in generating the questions, how was feedback gathered and integrated?

In summary I would suggest that their should be a more detailed description of what exactly is meant by participatory research and how it took place and a more detailed description also be given on how the interview guide and survey questions were devised.

Results

I like this section as it presents the data in a clear manner. On line 163 a Table 3 is mentioned the description of its content does not correspond to the Table 3 included in the manuscript. Hence this table appears to be missing. In line 301, the word “shitty” is a participant comment and should be in italics. It may be helpful to include a table with the main themes and the subthemes.

Discussion

This is good and supported by the results.

Reviewer #2: A very interesting and useful topic. However, the language standard across the board requires significant improvement, specifically around sentence structure, grammar, and in-text citation style. This paper is also quite undercited - would have done good to look at some of the work being done by Damien Riggs, Gavi Ansara, Shoshana Rosenberg, JR Latham and others. The arguments are not very well-supported by the literature provided, and lack nuance and critical assessment. There is also lack of clarity around a lot of the language being used e.g. "transfemmes", which might be somewhat common nomenclature in community but does not provide adequate description of what group of people is being addressed.

From a methodology standpoint, I'm not convinced 60 participants is enough for a quantitative component. Certainly it is a very limited number in terms of appropriate levels of power and validity. This section feels like it does not necessarily help to support your argument, at least not in its current form.

Having said all that, there is some fantastic data in here that would be very interesting if it underwent some further analysis and was sufficiently linked to current literature. Very interesting to talk about energy drain as a concept, something that requires more research and attention, particularly as we see more and more researchers engage the trans community for labour in the form of consultancy. I would be interested to see the qualitative interview protocol in particular, as it still seems like the questions could have been more nuanced and yielded some more in-depth data than what is presented here.

I believe this paper has some potential, but needs to be a) reviewed by a more senior academic for grammar, format, and syntax, and b) reconsider the methodological/analytical focus. While there's always some need for further data on discrimination, a 60-person survey does little to support your standpoint. You may be better off turning this into a purely qualitative analysis paper using the interview data on its own, particularly as the quant data points do not appear to be linked to the topic of the paper overall.

6. PLOS authors have the option to publish the peer review history of their article (what does this mean?). If published, this will include your full peer review and any attached files.

Reviewer #1: No

Reviewer #2: No

---

## [Author Response · Author response to Decision Letter 0]

3 Jun 2020

1. Is the manuscript technically sound, and do the data support the conclusions?

Reviewer #1: Yes

Reviewer #2: Partly

2. Has the statistical analysis been performed appropriately and rigorously?

Reviewer #1: Yes

Reviewer #2: Yes

We appreciate the reviewers’ positive commentary on our statistical analyses.

3. Have the authors made all data underlying the findings in their manuscript fully available?

Reviewer #1: No

Reviewer #2: Yes

Though this data comes from a larger dataset, all relevant excerpts are available at https://www.openicpsr.org/openicpsr/project/117143/version/V1/view. 

4. Is the manuscript presented in an intelligible fashion and written in standard English?

Reviewer #1: Yes

Reviewer #2: No

We have conducted a thorough spelling and grammar review of the manuscript and corrected mistakes.

5. Review Comments to the Author

Reviewer #1: Comments to author

The abstract summarizes the study in a comprehensive manner.

Introduction:

Good review of the pertinent literature.

We appreciate the reviewer’s positive comment of our literature review.

Line 40 I would remove “to be” after “remains”

We have corrected this as suggested.

The sentence starting by “Self-esteem” on line 76 presents pertinent information but may be perceived as too binary in its orientation. The sentence recounts the results from a study performed in 2018 and speaks to lack of familial acceptance decreasing the chance of completing sex reassignment surgery. Not all gender diverse individuals choose to undergo surgery and in fact many don’t want it. Therefore, completing surgery is not a required part of the self definition of one’s gender. The sentence appears to indicate the contrary. The authors may want to rethink how they present this data.

We appreciate the reviewer pointing out this oversight. We have changed the sentence as follows, “Self-esteem was also shown to be linked to familial acceptance, with lower familial acceptance associated with, for example, a decreased chance of future gender affirmation medical procedures for those individuals desiring them.”

Material and Methods

In my opinion this is the part of the manuscript that needs the most work. The authors have used a mixed method approach. The quantitative portion of the study uses a questionnaire that was devised de novo and which has not been validated. The sentence starting by “The initial” on line 119 briefly states that the interview guide and survey were informed by “the community partner”. This is all that is said about how these two instruments were developed. We need more information on that process. E.g. By whom and how were the questions used in the interview guide and survey originally drafted? Was this based on existing literature? What does “informed by the community partner” mean exactly? Were only one or two people consulted, was it a committee? How were these individuals chosen to give their opinion? Did they suggest questions, some or all? Did they read and say, yes that sounds fine”

We agree that this detail was an important omission from the methods section. We have revised this section and added the following, “The interview guide and survey was developed by study authors JB and TL, based on existing literature and preliminary conversations with the community partner, a GD support organization. The community partner’s leadership (two individuals) reviewed the guide and survey and made edits.”

As stated in line 127 and 128 the interviews occurred at a LGBTQ+ community organization. This may have led to individuals not wanting to participate as they may not have wanted to be seen entering such an area. This is not mentioned in the limitations and it should.

Upon review, we have added information about two of the interviews that were conducted over the phone instead of at the community organization. “Interviews took place in private rooms within an LGBTQ+ community organization or on the phone for those in extenuating circumstances (e.g. lacking transportation; n=2).” In the limitations section, we have added this as a limitation. “Because all in-person interviews were conducted at the community partner location, this may have excluded the experiences of those uncomfortable with the organization.”

The authors present a good description of their coding process, but some items need to be clarified. In Line 136 it is mentioned that “Each interview was coded twice by a trained coder”. This is not clear to me. Does this mean that the interview was coded by two different coders? Or twice by the same coder? As it is written it seems to be the latter. Is this right? 

We have revised this language for clarity to read, “Two different trained coders coded each interview such that each interview was coded twice.”

They also mention in line 138 that they presented the initial analyses to the community partner for feedback. How as this done, to how many people, where they the same as those that participated in generating the questions, how was feedback gathered and integrated?

We have included additional detail in this section as “Initial analyses were presented as a summary report to the community partner’s leadership for feedback in an in-person meeting.”

In summary I would suggest that their should be a more detailed description of what exactly is meant by participatory research and how it took place and a more detailed description also be given on how the interview guide and survey questions were devised.

We appreciate the reviewer’s request for more detail on what participation looked like, and we have addressed the aforementioned comments, as indicated above, to present these details in the revised manuscript.

Results

I like this section as it presents the data in a clear manner. On line 163 a Table 3 is mentioned the description of its content does not correspond to the Table 3 included in the manuscript. Hence this table appears to be missing.

We have corrected this to refer to the right table and thank the reviewer for drawing our attention to this mistake.

 In line 301, the word “shitty” is a participant comment and should be in italics.

We have put the word in italics as suggested.

 It may be helpful to include a table with the main themes and the subthemes.

We have added in Table 2 with the main themes and subthemes.

Discussion

This is good and supported by the results.

 We appreciate the reviewer’s compliment.

Reviewer #2: A very interesting and useful topic. 

We thank the reviewer for pointing this out. We agree that more strengths-based approaches are needed.

However, the language standard across the board requires significant improvement, specifically around sentence structure, grammar, and in-text citation style. 

We have conducted a thorough grammar and spelling review and corrected language as needed. However, we, of course, cannot correct language from direct quotations transcribed from the interview participants. We used Vancouver citation style, and all of our in-text citations appear to be in the correct format, as per PLOS One guidance to authors.

This paper is also quite undercited - would have done good to look at some of the work being done by Damien Riggs, Gavi Ansara, Shoshana Rosenberg, JR Latham and others. The arguments are not very well-supported by the literature provided, and lack nuance and critical assessment. 

We thank the reviewer for drawing our attention to these authors. We have added the following to the Introduction: “Another framework to describe the mental health effects of marginalization of trans people is cisgenderism which “delegitimizes people’s understanding of their genders and bodies” [4, 5].” And in the Discussion, “This relationship between discrimination and mental health is reported in other samples of GD individuals [5, 38].” We have also added the following to augment our assessment in the Discussion section: “Further, Case and Hunter [37] describe the power of “counterspaces” in supporting positive adaptation for marginalized individuals by “challenging deficit-oriented narratives.” We find evidence of all of the counterspace framework domains: narrative identity work in which participants described learning about others’ identities, acts of resistance in which participants described the value of freedom of expression that are not sanctioned in larger society, and direct relational transactions in which individuals offer social support, safety, and security.”

There is also lack of clarity around a lot of the language being used e.g. "transfemmes", which might be somewhat common nomenclature in community but does not provide adequate description of what group of people is being addressed.

We have added language to describe a meaning of transfemme as follows, “Pflum's [7] study of transfemme individuals (those who were not assigned female at birth and present as women or feminine, see [22] for more on this identity)…”

From a methodology standpoint, I'm not convinced 60 participants is enough for a quantitative component. Certainly it is a very limited number in terms of appropriate levels of power and validity.

Sampling from the trans/genderqueer/etc. population is oftentimes logistically complex and financially costly versus sampling from the overall population, with the additional complexity of limited literature to inform sample sizes a priori. There is no ‘minimal clinical importance difference’ to be defined in the context of the impact of discrimination on health and to inform an effect size. The hypotheses in this manuscript were not based on thresholds of minimal importance, but instead on whether there was any association between discrimination and self-assessed measures of health. Setting such a non-zero threshold would be ethically questionable, as it would classify some levels of the impact of discrimination on health as no evidence of such impact.

Sixty participants were sufficient to detect a significant, negative association between discrimination and perceived mental health (p<0.0001). There was some weak evidence of a potential negative association with physical health (p=0.159), as well. However, the sample size was, as mentioned in the manuscript, insufficient for the physical health measure given the observed variability to reach similar conclusions (if there is, indeed, such association, which of course we cannot claim).

Ultimately, this quantitative subsection can inform the reader and future researchers. Removing these findings from the manuscript would be detrimental for several reasons:

a) There is a statistically significant negative association between discrimination and self-assessed mental health among the population of focus in our study. This is an important finding that complements the existing literature and reinforces the need for health-driven action in this area, in particular for this highly-discriminated population. The evidence was overwhelming (p<0.0001) even with a reduced sample size. P-values have an embedded adjustment for sample sizes (usually by a factor of sqrt(n)). Smaller samples, therefore, require even further evidence to demonstrate statistical significance. 

b) Even though significant evidence of an association between discrimination and self-assessed physical health was not found (if it exists), providing these findings in a quantitative fashion can serve to power future studies. 

c) Non-significant findings across multiple independent studies can result in significant findings from a meta-analytic standpoint. Meta-analyses rely on usage of all studies, not only those with significant findings.

d) Ultimately, it is of ethical relevance to report all results, regardless of significance. Reporting results based on whether statistical significance is found would lead to p-hacking/selective reporting, which is a risk that would hurt disproportionally the under-researched populations or those suffering more subtle impacts. 

 This section feels like it does not necessarily help to support your argument, at least not in its current form.

We have added language to tie the quantitative component with the qualitative results:

In the Introduction: “Discrimination has been found to affect health among GD individuals, both in quantitative (including mental health [6-11] and risk behaviors [12, 13]) as well as qualitative studies (including mental health [14, 15] and health care access [16]).” And, “We use survey data to examine the effects of discrimination and interview data to examine the effects of GD community participation.”

In the Discussion: “Within the qualitative results, we see participants more focused on mental health impacts than physical health; the one subtheme that connected with physical health was that participants had difficulty finding resources (including healthcare providers) when they were less connected to the GD community.”

Having said all that, there is some fantastic data in here that would be very interesting if it underwent some further analysis and was sufficiently linked to current literature. Very interesting to talk about energy drain as a concept, something that requires more research and attention, particularly as we see more and more researchers engage the trans community for labour in the form of consultancy. 

We thank the reviewer for the compliment and agree that there are several very interesting ideas that come up within this work.

I would be interested to see the qualitative interview protocol in particular, as it still seems like the questions could have been more nuanced and yielded some more in-depth data than what is presented here.

This manuscript focuses on one segment of the interview, while we have others in press and under review that address other components of the study. We have added two example questions relevant to this manuscript’s analyses, “Example questions included, “What are the effects on your daily life when you participate in gender diverse groups/services?” and “What are the effects on your daily life when you don’t participate in gender diverse groups/services?”” 

I believe this paper has some potential, but needs to be a) reviewed by a more senior academic for grammar, format, and syntax, and b) reconsider the methodological/analytical focus. While there's always some need for further data on discrimination, a 60-person survey does little to support your standpoint. You may be better off turning this into a purely qualitative analysis paper using the interview data on its own, particularly as the quant data points do not appear to be linked to the topic of the paper overall.

Since this is a summary of prior comments, please note that we have addressed these issues throughout earlier comments provided (see above responses).

---

## [Decision Letter · Decision Letter 1]

3 Aug 2020

PONE-D-20-00790R1

“It just feels right”: Perceptions of the effects of community connectedness among gender diverse individuals

PLOS ONE

Dear Dr. Bowling,

Thank you for submitting your manuscript to PLOS ONE. After careful consideration, we feel that it has merit but does not fully meet PLOS ONE’s publication criteria as it currently stands. Therefore, we invite you to submit a revised version of the manuscript that addresses the points raised during the review process.

Thank you for your updated submission, which is nearly ready for acceptance. Please see the reviewer's comments below regarding updating and consistency of language with regard to the subjects of your study. Once those minor revisions are complete, we look forward to accepting your article.

We look forward to receiving your revised manuscript.

Kind regards,

Amy Michelle DeBaets, PhD

Academic Editor

PLOS ONE

Reviewers' comments:

Reviewer's Responses to Questions

**Comments to the Author**

1. If the authors have adequately addressed your comments raised in a previous round of review and you feel that this manuscript is now acceptable for publication, you may indicate that here to bypass the “Comments to the Author” section, enter your conflict of interest statement in the “Confidential to Editor” section, and submit your "Accept" recommendation.

Reviewer #2: (No Response)

2. Is the manuscript technically sound, and do the data support the conclusions?

Reviewer #2: Yes

3. Has the statistical analysis been performed appropriately and rigorously? 

Reviewer #2: Yes

4. Have the authors made all data underlying the findings in their manuscript fully available?

Reviewer #2: Yes

5. Is the manuscript presented in an intelligible fashion and written in standard English?

Reviewer #2: Yes

6. Review Comments to the Author

Reviewer #2: Some great improvements on this paper. My remaining concerns are about the use of GD rather than trans or TGD as the overall population descriptor. GD seems to be slowly falling out of favour, and certainly in the realm of trans research overall, "trans"/"transgender" is the most appropriate title used when discussing these communities. I also noticed that "gender nonbinary" is still being used, which is generally a misnomer. "Non-binary" (without the word "gender" in front) is the most accepted gender descriptor in the majority of research in the area. So the first line of the abstract, for example, might read better as:

"Trans people (e.g. trans men and women, nonbinary and gender fluid people) are at higher risk for..."

Otherwise the paper is looking good, and once these little bits are fixed it should make for a good contribution.

7. PLOS authors have the option to publish the peer review history of their article (what does this mean?). If published, this will include your full peer review and any attached files.

Reviewer #2: No

---

## [Author Response · Author response to Decision Letter 1]

3 Aug 2020

Comments to the Author

Reviewer #2: Some great improvements on this paper. My remaining concerns are about the use of GD rather than trans or TGD as the overall population descriptor. GD seems to be slowly falling out of favour, and certainly in the realm of trans research overall, "trans"/"transgender" is the most appropriate title used when discussing these communities. I also noticed that "gender nonbinary" is still being used, which is generally a misnomer. "Non-binary" (without the word "gender" in front) is the most accepted gender descriptor in the majority of research in the area. So the first line of the abstract, for example, might read better as:

"Trans people (e.g. trans men and women, nonbinary and gender fluid people) are at higher risk for..."

Otherwise the paper is looking good, and once these little bits are fixed it should make for a good contribution.

We appreciate the reviewer’s assistance with pointing out language changes and evolution. We have updated “gender diverse” to “trans” throughout the paper, and we have removed “gender” in front of “gender nonbinary” in the parenthetical descriptions.

---

## [Editor Report · Decision Letter 2]

24 Sep 2020

“It just feels right”: Perceptions of the effects of community connectedness among trans individuals

PONE-D-20-00790R2

Dear Dr. Bowling,

We’re pleased to inform you that your manuscript has been judged scientifically suitable for publication and will be formally accepted for publication once it meets all outstanding technical requirements.

Kind regards,

Amy Michelle DeBaets, PhD

Academic Editor

PLOS ONE
---

## [Editor Report · Acceptance letter]

25 Sep 2020

PONE-D-20-00790R2 

“It just feels right”: Perceptions of the effects of community connectedness among trans individuals 

Dear Dr. Bowling:

I'm pleased to inform you that your manuscript has been deemed suitable for publication in PLOS ONE. Congratulations! Your manuscript is now with our production department. 

Kind regards, 

on behalf of

Dr. Amy Michelle DeBaets 

Academic Editor

PLOS ONE